# Proximal Preference Optimization for Diffusion Models

## Abstract

Preference optimization techniques such as Reinforcement Learning From Human/AI Feedback(RLHF/RLAIF) emerge as the new standard approach in fine-tuning foundation models. Preference learning, however, is often optimized under the reinforcement learning setting which leads to a high variance, low data efficiency, as well as much longer steps to converge. Recent study of Direct Preference Optimization proved to be an effective way to mitigate such issues by converting the preference learning into a supervised learning paradigm for language models. However, little have been studied in the case of image generative models such as diffusion models. In this paper, we propose Proximal Preference Optimization for Diffusion models (PPOD) that extends the prior work with proximal constraints to solve the optimization challenges in diffusion model. We further study the online vs offline evaluation as well as the optimization objective choices and figure out the optimal setting for different use cases. Such a method makes preference optimization stable and feasible under the supervised learning setting. Our evaluation shows PPOD outperforms the other RL based reward optimization approaches on the stable diffusion model. To the best of our knowledge, we are the first work that enabled the efficient optimization for the RLAIF on the diffusion models.

## 1 Introduction

Diffusion models (Ho et al., 2020; Song et al., 2020) have demonstrated impressive capabilities for generating high-quality images from random Gaussian noise going through multiple iterative denoising steps. Despite the stunning quality, it is difficult to control the generated content from randomly sampled noise. Recent large-scale text-to-image diffusion models, e.g. Imagen (Saharia et al., 2022), DALL-E 2 (Ramesh et al., 2022), and Stable Diffusion (SD) (Rombach et al., 2022) incorporate text encoders (e.g. CLIP (Radford et al., 2021) and T5 (Raffel et al., 2020)) to enable generating high-quality images from text prompts. The generated content becomes therefore more controllable under the text prompt guidance. However, all these approaches train diffusion models via supervised learning, which makes generated images follow the training data distribution. These models generally do not perform well in generating images for the properties that are not well learned from the training dataset, e.g. specific color, image composition, object count, object location, etc. Although supervised finetuning (SFT) can alleviate this problem, the model is very easy to over-fit on the content details, leaving the image-text alignment issue being unresolved.

Learning from human or AI feedback has been verified to be an effective solution to solve the limitation of supervised learning for generative models. Lee et al. (2023b) explored learning a reward model from human feedback, followed by SFT of the text-to-image model with a reward-weighted loss. Although their approach achieves better reward scores and improves the generation capabilities on some properties, e.g. specified color, object count, and background, the model still suffers from the limitations of SFT. A very recent work (Black et al., 2023) first scoped the denoising process in diffusion models as a multi-step decision making problem, which enabled using policy gradient algorithm namely denoising diffusion policy optimization (DDPO) to optimize the diffusion model. Their work demonstrated that DDPO is more effective than SFT, and is able to adapt text-to-image diffusion models to learn the difficult concepts that are hard to express by text prompts, such as image compressibility, aesthetic, etc. Another recent work (Fan et al., 2023) proposed using online reinforcement learning (RL) to finetune text-to-image models. Their approach integrated diffusion policy optimization from an online reward model with a KL regularization, namely DPOK. Their

experiments verified that RL can improve both the image-text alignment and image quality compared to basic SFT. Both DDPO and DPOK optimize diffusion models via the RL policy optimization approach, i.e. proximal policy optimization (PPO; Schulman et al., 2017).

In large language models, Rafailov et al. (2023) proposed direct preference optimization (DPO) to transform the RL with human preference to a supervised learning setting. The core idea of DPO is using the Bradley-Terry (BT; Bradley & Terry, 1952) model to model the human preference between a pair of sampled data. The optimization problem is therefore transformed from a reward optimization to a cross-entropy loss optimization problem. A big advantage of DPO is that it bypasses the explicit reward modeling step, and avoids the need to perform reinforcement learning optimization. Their studies showed that DPO is efficient and effective, since it achieved similar or even better performance than other policy optimization approaches such as proximal policy optimization (PPO) and trust region policy optimization (TRPO; Schulman et al., 2015) etc.

Inspired by the success of DPO on language models, we explore finetuning the diffusion model by the direct preference optimization. Since the human feedback is expensive and the AI evaluation performance achieves remarkable progress, the AI feedback is getting increasing attention. We therefore focus on optimizing the diffusion model by directly optimizing the reward from RLAIF. In our work, we choose ImageReward (Xu et al., 2023), a prelearned reward model that shows the best alignment with human evaluation among all the image quality reward models as the AI evaluator. We derive the optimization algorithm for diffusion models by considering the iterative denosing steps in the preference modeling. We also introduce the proximal principle featured in PPO into our approach to improve the model training stability. Some further studies about performance of optimizing the model with both online and offline preference data demonstrate the online optimization can achieve state-of-the-art (SOTA) performance for learning the properties in text prompts, e.g. specified color, image composition, object count, object location, etc. Figure 1 overviews optimization frameworks between the traditional RL based approach adopted by previous work (e.g. DDPO and DPOK) and the proposed PPOD approach.

The main contribution of our paper is proposing a novel reward maximization algorithm, **P**roximal **P**reference **O**ptimization for **D**iffusion models, namely **PPOD**, for RLAIF. We rederive the DPO algorithm for diffusion models with RLAIF, but find that it cannot work out of the box. Our investigation reveals that by further incorporating elements from PPO, such as the proximal update principle and online training, we are able to drastically improve the model training stability and performance. Our PPOD approach enables direct preference optimization for diffusion models for the first time and achieves superior performance than PPO-based methods.

## 2 RELATED WORK

**Text-to-image diffusion.** Diffusion models (Ho et al., 2020; Song & Ermon, 2019; Saharia et al., 2021b;a) have emerged as an effective class of approaches for high-quality image generation via an iterative denoising process to transform Gaussian noise into image samples.

The text-to-image generation approaches (Rombach et al., 2022; Ramesh et al., 2022; Ho et al., 2022; Dhariwal & Nichol, 2022; Nichol et al., 2021; Saharia et al., 2022) demonstrate remarkable performance on fine-grained high-resolution image generation conditioned by text prompts. Some text-to-image diffusion approaches (Rombach et al., 2022) replace the pixel space denoising with latent space denoising to significantly reduce the computational complexity. Since the denoising training is defined as a supervised learning process, the generated images follow the training data distribution.

Although text-to-image diffusion models introduce text encoders to condition the denoising process, from a user's perspective, generating images with desired content from text prompts is still not easy. To improve model adaption to user context, a number of recent work focus on fine-tuning the model with limited user data (Ruiz et al., 2023; Gal et al., 2023; Sohn et al., 2023; Kawar et al., 2023). Although fine-tuning can adapt the generated data distribution to the distribution learned from user input data, it still cannot learn the semantic properties in text prompts including specific color, image composition, object count, object location, etc. Fine-tuning the model to be able to generate unlearned properties in text prompts remains an open research problem.

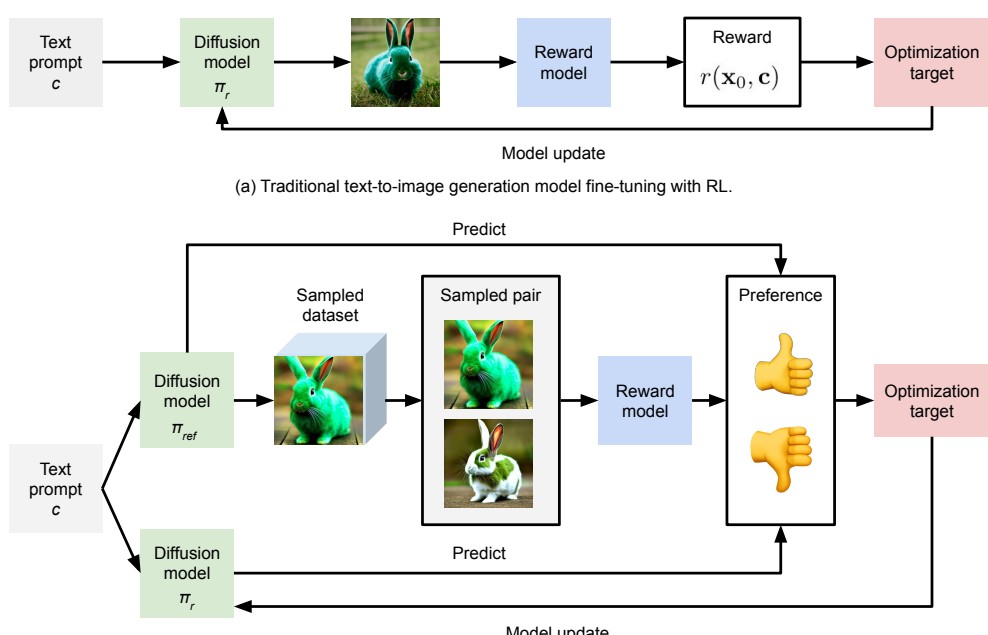

(a) Traditional text-to-image generation model fine-tuning with RL.

(b) Our proximal preference optimization for fine-tuning text-to-image generation model.

**Figure 1:** Comparison between (a) the traditional RL based diffusion model finetuning framework and (b) the proposed PPOD framework. The traditional finetuning approach (e.g. Fan et al. (2023)) performs an online RL by predicting a reward for each generated image and optimizing the KL-divergence regularized reward objective. Our approach converts the policy optimization to a supervised learning problem. It samples a pair of images from a generated image dataset (generated either online or offline). Predict the preference from the current and reference diffusion models. Finally optimize the current diffusion model via a cross-entropy optimization objectives.

**Learning from human / AI feedback.** Introducing human / AI assessment to improve the performance of generative models has become a popular approach, since the log likelihood optimization is not sufficient to optimize generative models. The reinforcement learning from human / AI feedback, i.e. (RLHF; Ouyang et al., 2022) and (RLAIF; Lee et al., 2023a), has been widely studied in language modeling. The human feedback is collected by asking the annotator to rank the outputs from the language model, and called preference data. A reward model is learned from the human preference data and used for the policy optimization. The main challenges of RLHF / RLAIF is the learning complexity. Schulman et al. (2017) proposed proximal policy optimization (PPO) to reduce the optimization complexity by clapping the gradient during the learning. Schulman et al. (2015) proposed the trusted region policy optimization (TRPO) to limit the exploration space within the trusted region. Recently, Rafailov et al. (2023) proposed a direct preference optimization (DPO), which transforms the reward optimization to a cross-entropy loss optimization problem by applying the BT model (Bradley & Terry, 1952) to model the preference data. Their evaluation demonstrated that DPO performs similarly or better than existing RLHF algorithms, and reduces the complexity by skipping training reward model.

**Reinforcement learning to fine-tune diffusion model.** Fine-tuning text-to-image generation models with RL is an emerging research topic given the success of RLHF / RLAIF in language modeling. The early work learned a reward model for image generator by fine-tuning the vision-language models such as CLIP (Radford et al., 2021) and BLIP (Li et al., 2022) to predict the scalar reward value that is expected to be aligned with the human preference (Xu et al., 2023; Kirstain et al., 2023). The reward model is then used to fine-tune the generative model (Wu et al., 2023; Lee et al., 2023b). Recently, Black et al. (2023) introduced an online RL learning pipeline by treating the denoising process as a Markov Decision Process (MDP) and solving it by PPO. In parallel, Fan et al. (2023) also proposed an online fine-tuning framework to optimize the reward function plus the KL divergence between the reference and fine-tuned diffusion model. These RL based fine-tuning approaches explored a new

direction of enabling diffusion models to learn under the guidance. The learning efficiency is not well optimized since they all required to build a reward model from the preference data. We develop a proximal preference optimization to directly optimize on the preference data. Evaluation results demonstrate that our approach outperforms existing RL based fine-tuning approaches for diffusion models.

## 3 METHOD

### 3.1 PROBLEM SETTING

**Diffusion model**. In our work, we use the stable diffusion (SD) model (Rombach et al., 2022) as the starting point for RL finetuning. The SD model is trained based on denoising diffusion probabilistic models (DDPMs; Ho et al., 2020). Let $\mathbf{x}_0$ be the original image, and $\mathbf{x}_T$ be a noise image sampled from a Gaussian distribution. The forward pass is a Markovian process $q(\mathbf{x}_t|\mathbf{x}_{t-1})$, which transforms the original image $\mathbf{x}_0$ to $\mathbf{x}_T$ by iteratively adding Gaussian noise, producing a sequence of noisy samples $\{\mathbf{x}_1, \cdots, \mathbf{x}_T\}$. The Markovian process can be described as:

$$q(\mathbf{x}_t|\mathbf{x}_{t-1}) = \mathcal{N}(\mathbf{x}_t; \sqrt{1-\beta_t}\mathbf{x}_{t-1}, \beta_t\mathbf{I}) , \quad q(\mathbf{x}_{1:T}|\mathbf{x}_0) = \prod_{t=1}^{T} q(\mathbf{x}_t|\mathbf{x}_{t-1}) . \quad (1)$$

The denoising process is trained to reverse the forward process to convert from a noise input $\mathbf{x}_T \sim \mathcal{N}(\mathbf{0}, \mathbf{I})$ to the original image $\mathbf{x}_0$ as below:

$$p_\theta(\mathbf{x}_{0:T}) = p(\mathbf{x}_T)\prod_{t=1}^{T} p_\theta(\mathbf{x}_{t-1}|\mathbf{x}_t) , \quad p_\theta(\mathbf{x}_{t-1}|\mathbf{x}_t) = \mathcal{N}(\mathbf{x}_{t-1}; \boldsymbol{\mu}_\theta(\mathbf{x}_t, t), \sigma_t^2\mathbf{I}) . \quad (2)$$

The distribution $p_\theta(\mathbf{x}_{t-1}|\mathbf{x}_t)$ can be modeled as a Gaussian when the noise at each step is small. Its mean is then predicted by the neural network $\boldsymbol{\mu}_\theta(\mathbf{x}_t, t)$ based on the input noisy image $\mathbf{x}_t$ at timestamp $t$.

**Text-to-image diffusion model**. Text-to-image diffusion models introduce text prompts to condition the image generation. The encoded text prompt is attended to the latent code during the denoising process. Therefore, the prediction network can be updated to $\boldsymbol{\mu}_\theta(\mathbf{x}_t, t, \mathbf{c})$ where $\mathbf{c}$ is the conditional text prompt.

**Reinforcement learning with reward model**. We introduce a reward model into the finetuning loop to guide the performance improvement. The reward model steers the finetuning process by evaluating the generated image quality and alignment with the text prompt. The reward function is defined as $r(\mathbf{x}_0, \mathbf{c})$. Since the denoising process is defined as a MDP (Black et al., 2023; Fan et al., 2023), which is formalized as a sequential decision making process, the diffusion model can be treated as a policy $\pi_\theta$. The diffusion model defines a conditional probability distribution $\pi_\theta(\mathbf{x}_0|\mathbf{c})$.

### 3.2 PROXIMAL PREFERENCE OPTIMIZATION

The proximal preference optimization for diffusion (PPOD) models is proposed to optimize the reward. The optimization problem of PPOD is defined as

$$\max_\pi \mathbb{E}_{\mathbf{c} \sim p(\mathbf{c})}\big[\mathbb{E}_{\mathbf{x}_0 \sim \pi(\mathbf{x}_0|\mathbf{c})}[r(\mathbf{x}_0, \mathbf{c})] - \eta\mathrm{KL}[\pi(\mathbf{x}_{0:T} \mid \mathbf{c}) \parallel \pi_{\mathrm{ref}}(\mathbf{x}_{0:T} \mid \mathbf{c})]\big] , \quad (3)$$

where $\pi_{\mathrm{ref}}(\mathbf{x}_0|\mathbf{c})$ is the reference policy before finetuning, i.e. the pretrained diffusion model. $\mathbb{E}_{\mathbf{x}_0 \sim \pi(\mathbf{x}_0|\mathbf{c})}[r(\mathbf{x}_0, \mathbf{c})]$ defines the objective to optimize reward. KL denotes the KL divergence between the finetuned and reference policies, which regularizes the finetuned policy to maintain the overall generation capability of the pretrained policy. Different from the language modeling, the KL divergence for diffusion models needs to regularize the whole denoising process between the finetuned and reference policies, i.e. $\pi(\mathbf{x}_{0:T} \mid \mathbf{c})$ and $\pi_{\mathrm{ref}}(\mathbf{x}_{0:T} \mid \mathbf{c})$. Finally, $\eta$ denotes a weighting coefficient between the reward and KL divergence regularization.

Inspired by the derivation in Rafailov et al. (2023), the optimal solution to Equation (3) can be expressed as

$$\pi_r(\mathbf{x}_{0:T} \mid \mathbf{c}) = \frac{1}{Z(\mathbf{c})}\pi_{\mathrm{ref}}(\mathbf{x}_{0:T} \mid \mathbf{c})\exp\left(\frac{1}{\eta}r(\mathbf{x}_0, \mathbf{c})\right) , \quad (4)$$

**Training without Clipped Log Ratios**

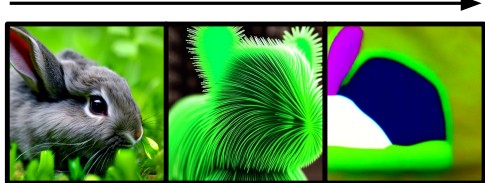

**Training with Clipped Log Ratios**

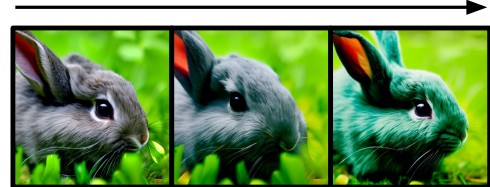

**Figure 2: Effect of clipping log probability ratios during training.** Clipping log probability ratios remarkably improves optimization stability.

where the denominator $Z(\mathbf{c})$ can be written as

$$Z(\mathbf{c}) = \int \pi_{\text{ref}}(\mathbf{x}_{0:T} \mid \mathbf{c}) \exp\left(\frac{1}{\eta} r(\mathbf{x}_0, \mathbf{c})\right) \mathrm{d}\mathbf{x}_{0:T} . \tag{5}$$

Predicting the optimal policy $\pi_r$ from Equation (4) is hard, since the denominator $Z(\mathbf{c})$ is intractable. However, the Bradley-Terry (BT; Bradley & Terry, 1952) model designed for modeling the preference data inspires us to solve the problem via modeling the reward difference between samples. Assume we have a pair of generated image samples $\{\mathbf{x}_0^w, \mathbf{x}_0^l\}$, and $\mathbf{x}_0^w$ achieves higher reward value than $\mathbf{x}_0^l$. The BT model can therefore be framed as optimizing the following cross-entropy loss with a parameterized reward model $r(\mathbf{x}_0, \mathbf{c})$

$$\max_r \ \mathbb{E}_{(\mathbf{x}_0^w, \mathbf{x}_0^l, \mathbf{c}) \sim \mathcal{D}} \left[ \log \sigma\left(r(\mathbf{x}_0^w, \mathbf{c}) - r(\mathbf{x}_0^l, \mathbf{c})\right)\right] , \tag{6}$$

where $\sigma$ is the logistic function. Besides the Sigmoid function, we can also consider the other loss functions such as MSE, etc. We evaluate the performance between cross-entropy and MSE loss in the ablation studies.

By transforming Equation (4), we can express the reward for each sample as

$$r(\mathbf{x}_0, \mathbf{c}) = \eta \log \frac{\pi_r(\mathbf{x}_{0:T} \mid \mathbf{c})}{\pi_{\text{ref}}(\mathbf{x}_{0:T} \mid \mathbf{c})} + \eta \log Z(\mathbf{c}) . \tag{7}$$

Substituting $r(\mathbf{x}_0, \mathbf{c})$ defined in Equation (7) in Equation (6) yields the following optimization target

$$\max_{\pi_r} \ \mathbb{E}_{(\mathbf{x}_{0:T}^w, \mathbf{x}_{0:T}^l, \mathbf{c}) \sim \mathcal{D}} \left[ \log \sigma\left(\eta \log \frac{\pi_r(\mathbf{x}_{0:T}^w \mid \mathbf{c})}{\pi_{\text{ref}}(\mathbf{x}_{0:T}^w \mid \mathbf{c})} - \eta \log \frac{\pi_r(\mathbf{x}_{0:T}^l \mid \mathbf{c})}{\pi_{\text{ref}}(\mathbf{x}_{0:T}^l \mid \mathbf{c})}\right)\right] . \tag{8}$$

Now, the denominator $Z(\mathbf{c})$ is canceled out in Equation (8), which makes the optimization problem solvable. We can interpret Equation (8) as using the difference between two log probability ratios to predict which generated image is preferred by humans or a reward model. That is, the policy is optimized for predicting the preference, instead of directly maximizing the reward.

Since the diffusion model can be treated as a MDP, $\mathbf{x}_{0:T}$ can be expanded as a product of the step-wise denoising step. The log ratio term in Equation (8) can be transformed to a sum of the step-wise log ratios as:

$$\log \frac{\pi_r(\mathbf{x}_{0:T}^w \mid \mathbf{c})}{\pi_{\text{ref}}(\mathbf{x}_{0:T}^w \mid \mathbf{c})} = \sum_{t=1}^{T} \left( \log \frac{\pi_r(\mathbf{x}_{t-1}^w \mid \mathbf{x}_t^w, \mathbf{c})}{\pi_{\text{ref}}(\mathbf{x}_{t-1}^w \mid \mathbf{x}_t^w, \mathbf{c})} \right) . \tag{9}$$

Our experiments found optimizing Equation (8) without any regularization can produce very unstable fine-tuing results as demonstated in the left sub-figure of Figure 2. To improve the stability of the optimization process, we are inspired by the PPO approach to clip the log ratio between the finetuned and reference policies by $\text{clog}(\cdot) = \text{clip}\left(\log(\cdot), -\epsilon, \epsilon\right)$. After introducing the clip, the optimization output becomes much more stable than without clipping. The images in the right sub-figure of Figure 2 demonstrate the tractable progress of the optimization process.

Therefore, the optimization targets in Equation (8) can be further simplified as

$$\max_{\pi_r} \ \mathbb{E}_{(\mathbf{x}_{0:T}^w, \mathbf{x}_{0:T}^l, \mathbf{c}) \sim \mathcal{D}} \left[ \log \sigma\left(\eta \sum_{t=1}^{T} \left( \text{clog} \frac{\pi_r(\mathbf{x}_{t-1}^w \mid \mathbf{x}_t^w, \mathbf{c})}{\pi_{\text{ref}}(\mathbf{x}_{t-1}^w \mid \mathbf{x}_t^w, \mathbf{c})} - \text{clog} \frac{\pi_r(\mathbf{x}_{t-1}^l \mid \mathbf{x}_t^l, \mathbf{c})}{\pi_{\text{ref}}(\mathbf{x}_{t-1}^l \mid \mathbf{x}_t^l, \mathbf{c})} \right)\right)\right] . \tag{10}$$

---

**Algorithm 1** PPOD Training

---

**Require:** A pretrained denoising model $\pi_{\text{ref}}(\mathbf{x}_{0:T} \mid \mathbf{c})$, a reward model $r(\mathbf{x}_0, \mathbf{c})$
  **while** not converged **do**
    Sample $K$ images from the given text prompt $c$ and record the sampling process $(\mathbf{x}_{0:T}^k \mid \mathbf{c})$.
    Predict a reward score $r(\mathbf{x}^k, \mathbf{c})$ for each sampled image $\mathbf{x}_k$.
    Sample $M$ pairs of images from the rated images generated last step.
    Calculate the preference prediction loss defined in Equation (10).
    Optimize the model with gradient descent.
  **end while**

---

The PPOD algorithm is outlined in Algorithm 1. PPOD can optimize with the reward generated both online and offline. The offline generated reward is used in RLHF when the human preference data is collected offline. In this case, the preference pair are sampled from the offline dataset in each batch to optimize the model. On the other side, the online generated reward can be used in RLAIF or interactive finetuning. In this case, a new dataset is generated at each training step from the latest finetuned generator. The reward model or human annotator rates the sampled pair as the preference for each iteration. Based on our evaluation, the online learning can achieve better performance than the offline learning.

## 4 EXPERIMENTS

### 4.1 EXPERIMENTAL SETUP

We follow the same setting as DPOK (Fan et al., 2023), where we use Stable Diffusion v1.5 (Rombach et al., 2022) as the pretrained model, and finetune it on one prompt at a time. Specifically, we use the following prompts for training: "A green colored rabbit", "Four wolves in the park", "A cat and a dog", and "A dog on the moon". They involve generating objects with specific colors, counts, compositions, and locations, which have been found challenging for Stable Diffusion. It has been shown in DPOK that by finetuning Stable Diffusion through reward maximization, we can partially mitigate these weaknesses. Our results will demonstrate that our proposed model PPOD obtains a better reward than DPOK on all four prompts, showing great potential for RLAIF in diffusion models.

**Baselines.** In addition to the pretrained Stable Diffusion, we mainly consider supervised finetuning (SFT) and DPOK to be our baselines. Because the source code of DPOK has not been released, we follow the training setup described in the DPOK paper. Specifically, we use ImageReward (Xu et al., 2023) as the reward model, as it is better aligned with human preference than alternatives such as CLIP (Radford et al., 2021). The number of queries to the reward model is 20K for all methods compared. That is, SFT generates an offline dataset of 20K images using the pretrained Stable Diffusion. DPOK generates 20K images in total during its online policy update. For our model, the offline version uses the same amount of data as SFT. The online version consists of 4 stages. For each stage, we generate 5K images using the latest model checkpoint, and update the model till convergence on these 5K images.

### 4.2 MAIN RESULTS

We report the ImageReward score of all models in Figure 3. The scores of Stable Diffusion and PPOD are averaged over 5K samples per prompt using the same random seed. The scores of SFT and DPOK are obtained from the DPOK paper, which uses 50 samples. PPOD achieves the best ImageReward score across all four prompts, surpassing the PPO-based DPOK and significantly outperforming SFT. This suggests that optimizing policy for preference prediction is a promising alternative to direct reward maximization for RLAIF in diffusion models.

In Figure 4 (Right), we show PPOD generation samples at the end of each online training stage. It can be seen that PPOD is able to continually improve through online training, correcting mistakes in the pretrained Stable Diffusion and previous training stages. For example, in the second row, PPOD gradually increases the number of wolves to four. In the third row, PPOD removes the additional cat

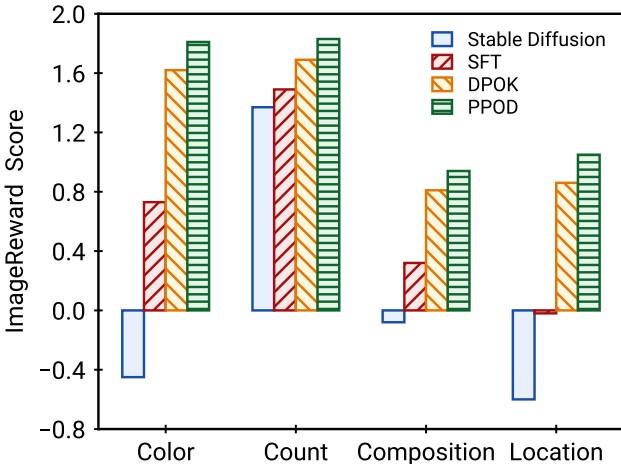

**Figure 3: Comparison of reward maximization capability.** Our proposed model PPOD achieves the best ImageReward score across all four prompts, showing great potential for RLAIF.

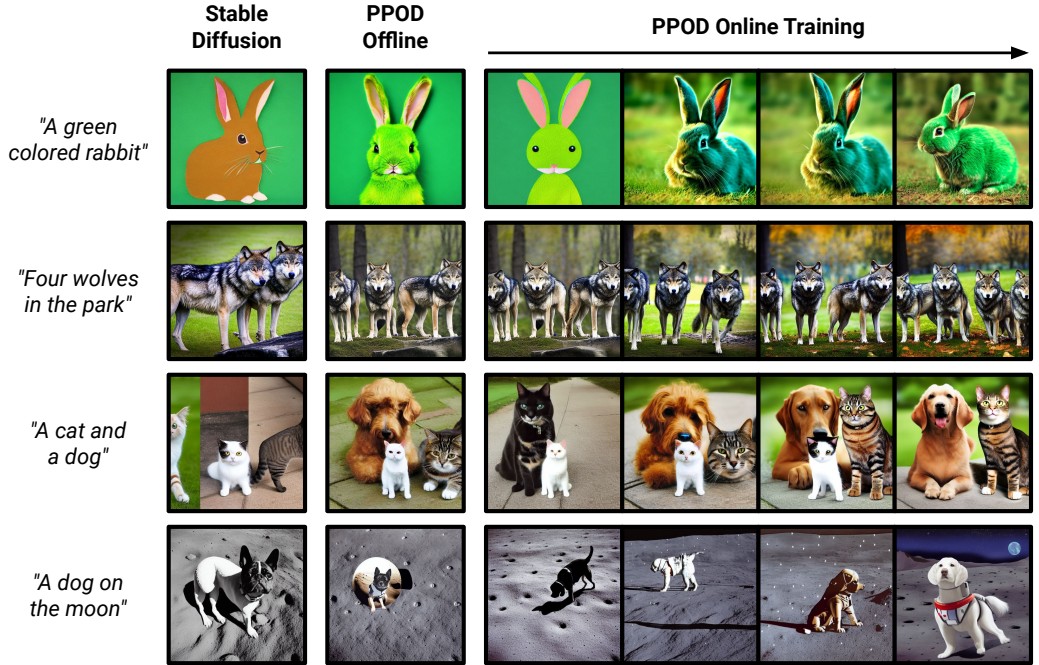

**Figure 4: Generation samples on training prompts.** The first image on each row is the image generated by Stable Diffusion. It shows limitations in generating the correct color, count, composition, and location. The second image on each row is generated by the PPOD offline version. It can partially address the limitations of the pretrained SD model. The last four images in a row on each row are the images generated by PPOD at different online training stages. Through online training, PPOD is able to continually improve its generation and correct mistakes. Images in the same row are generated with the same random seed.

in the last training stage. Because there is no DPOK checkpoint available, we are unable to provide a qualitative comparison with DPOK. We refer the reader to the DPOK paper for qualitative results.

We also test PPOD with unseen prompts at the end of each online training stage. Generation samples are shown in Figure 5 (Right). We find that PPOD generalizes reasonably, and the generation quality on unseen prompts also improves in the process of online training.

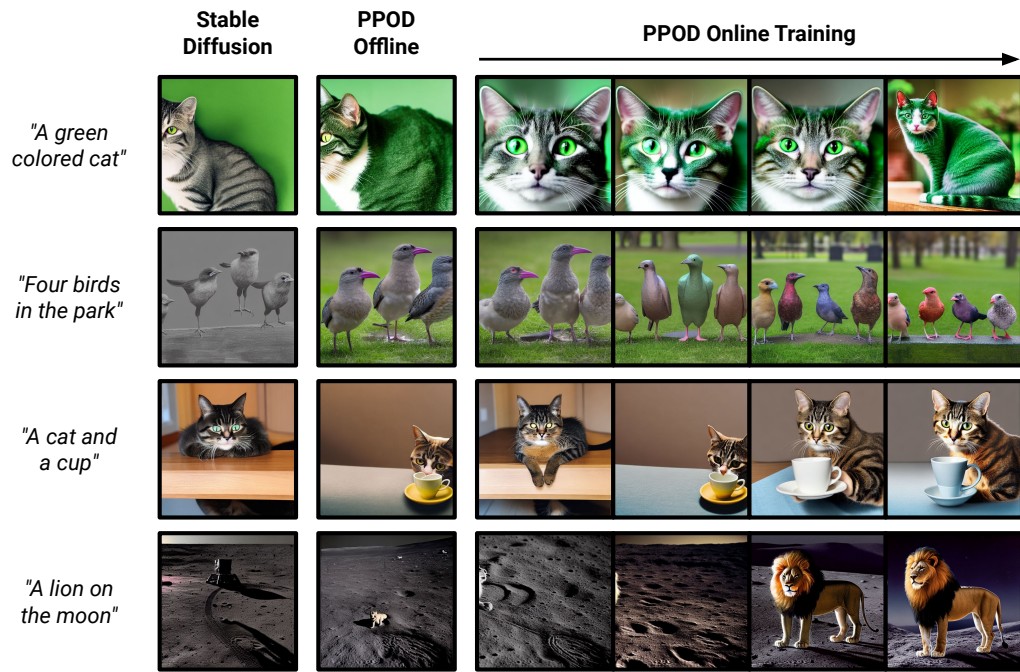

**Figure 5: Generation samples on unseen prompts.** PPOD maintains reasonable generalization ability. The generation quality on unseen prompts also improves over iterations in online training. Images in the same row are generated with the same random seed.

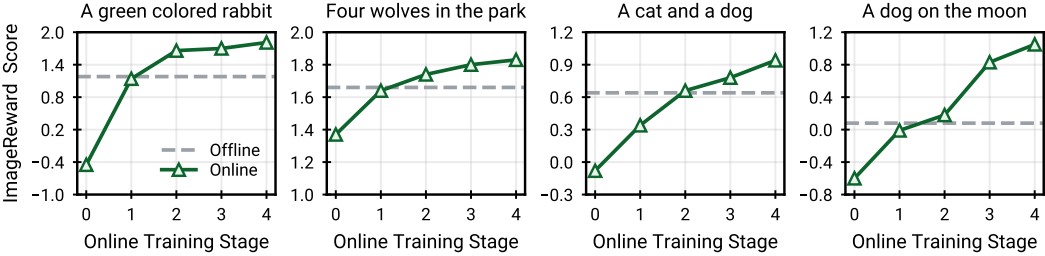

**Figure 6: Online vs. offline learning on training prompts.** The online model matches the offline model performance within two stages, and keeps improving afterwards in most cases.

### 4.3 ABLATION STUDY

**Online Learning vs. Offline Learning.** In Figures 4 and 5, we have qualitatively demonstrated that online training continually improves the generation quality of PPOD on both training prompts and unseen prompts. We further provide quantitative comparisons in Figures 6 and 7. Here, we evaluate the ImageReward score of PPOD at the end of each online training stage, for both training and unseen prompts. For a fair comparison, both the offline and online model use the same total number of reward queries and gradient updates. We find that in most cases, the online model is able to match the offline model performance within two stages (using half the number of reward queries and gradient updates), and keeps improving afterwards. We conjecture that with more frequent online updates, PPOD may become even better.

**Preference Prediction Loss: Cross Entropy vs. Mean Squared Error.** Here, we compare PPOD with its variant, PPOD-MSE, that uses mean squared error (MSE) as the preference prediction loss. For simplicity, we only consider the offline setting. We observe in Table 1 that the two models obtain similar ImageReward scores. Qualitatively, as shown in Figure 8, PPOD-MSE generations can sometimes be less natural. Therefore, we choose cross-entropy as the favorable loss function in our design.

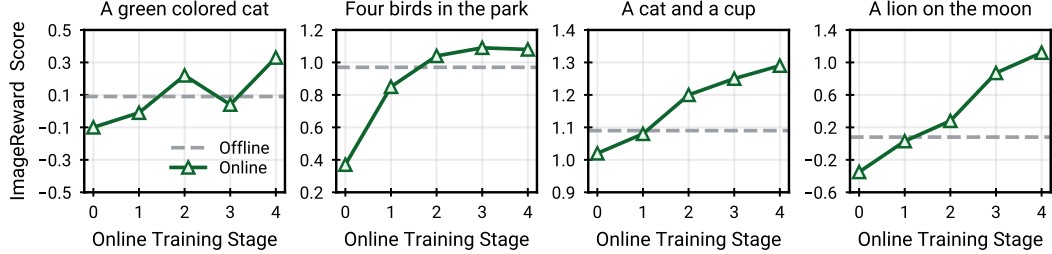

**Figure 7: Online vs. offline learning on unseen prompts.** The online model matches the offline model performance within two stages, and keeps improving afterwards in most cases.



**PPOD Offline**  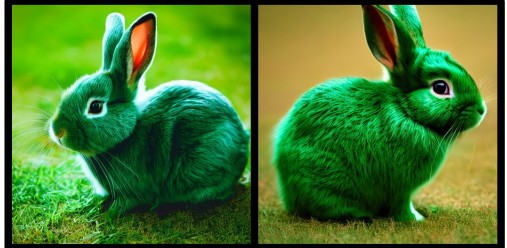  **PPOD-MSE Offline**  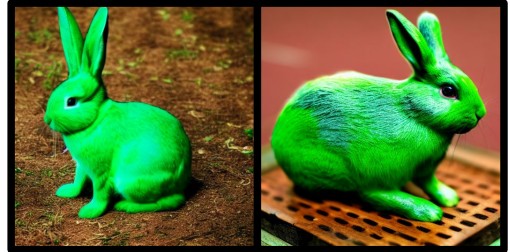



**Figure 8:** Generated images learned with cross-entropy and MSE as the preference prediction loss, respectively.

## 5 CONCLUSION

In this paper, we propose a novel proximal preference optimization approach to fine-tune the diffusion model (PPOD) for the use case of reinforcement learning with AI feedback (RLAIF). PPOD derives the direct preference optimization algorithm for diffusion process and introduces the proximal constraint to solve the optimization challenges in diffusion model. Similar to DPO, PPOD reduces the barriers in fine-tuning the diffusion model and makes the fine-tuning process efficient and stable. Our experiments demonstrate the superior performance for maximizing the reward model with fine-tuning the diffusion model, over both the supervised fine-tuning and the RL based approach. We also study the design including online and offline learning as well as the supervised learning loss choice. The evaluation proves our design achieves the optimal performance. We mainly target on RLAIF in this paper as RLAIF is getting more attention due to its low cost and comparable performance to RLHF. Notably, the proposed PPOD is a generic approach for various use cases including online RLAIF, iterative RLHF, the offline RLHF, etc.

## REFERENCES

Kevin Black, Michael Janner, Yilun Du, Ilya Kostrikov, and Sergey Levine. Training diffusion models with reinforcement learning. *arXiv preprint arXiv:2305.13301*, 2023.

**Table 1:** Quantitative evaluation of using cross-entropy and MSE as the preference prediction loss.

| Model | PPOD Offline | PPOD-MSE Offline |
|---|---|---|
| ImageReward Score | 1.18 | 1.15 |

Ralph Allan Bradley and Milton E Terry. Rank analysis of incomplete block designs: I. the method of paired comparisons. *Biometrika*, 39(3/4):324–345, 1952.

Prafulla Dhariwal and Alex Nichol. Diffusion models beat gans on image synthesis. In *Conference on Neural Information Processing Systems*, 2022.

Ying Fan, Olivia Watkins, Yuqing Du, Hao Liu, Moonkyung Ryu, Craig Boutilier, Pieter Abbeel, Mohammad Ghavamzadeh, Kangwook Lee, and Kimin Lee. Dpok: Reinforcement learning for fine-tuning text-to-image diffusion models. *arXiv preprint arXiv:2305.16381*, 2023.

Rinon Gal, Yuval Alaluf, Yuval Atzmon, Or Patashnik, Amit Haim Bermano, Gal Chechik, and Daniel Cohen-or. An image is worth one word: Personalizing text-to-image generation using textual inversion. In *International Conference on Learning Representations*, 2023.

Jonathan Ho, Ajay Jain, and Pieter Abbeel. Denoising diffusion probabilistic models. *Advances in Neural Information Processing Systems*, 33:6840–6851, 2020.

Jonathan Ho, Chitwan Saharia, William Chan, David J Fleet, Mohammad Norouzi, and Tim Salimans. Cascaded diffusion models for high fidelity image generation. *Journal of Machine Learning Research*, 2022.

Bahjat Kawar, Shiran Zada, Oran Lang, Omer Tov, Huiwen Chang, Tali Dekel, Inbar Mosseri, and Michal Irani. Imagic: Text-based real image editing with diffusion models. In *Proceedings of the IEEE/CVF Conference on Computer Vision and Pattern Recognition*, pp. 6007–6017, 2023.

Yuval Kirstain, Adam Polyak, Uriel Singer, Shahbuland Matiana, Joe Penna, and Omer Levy. Pick-a-pic: An open dataset of user preferences for text-to-image generation. *arXiv preprint arXiv:2305.01569*, 2023.

Harrison Lee, Samrat Phatale, Hassan Mansoor, Kellie Lu, Thomas Mesnard, Colton Bishop, Victor Carbune, and Abhinav Rastogi. Rlaif: Scaling reinforcement learning from human feedback with ai feedback. *arXiv preprint arXiv:2309.00267*, 2023a.

Kimin Lee, Hao Liu, Moonkyung Ryu, Olivia Watkins, Yuqing Du, Craig Boutilier, Pieter Abbeel, Mohammad Ghavamzadeh, and Shixiang Shane Gu. Aligning text-to-image models using human feedback. *arXiv preprint arXiv:2302.12192*, 2023b.

Junnan Li, Dongxu Li, Caiming Xiong, and Steven Hoi. Blip: Bootstrapping language-image pre-training for unified vision-language understanding and generation. In *International Conference on Machine Learning*, pp. 12888–12900. PMLR, 2022.

Alex Nichol, Prafulla Dhariwal, Aditya Ramesh, Pranav Shyam, Bob McGrew Pamela Mishkin, Ilya Sutskever, and Mark Chen. Glide: Towards photorealistic image generation and editing with text-guided diffusion models. *arXiv preprint arXiv:2112.10741*, 2021.

Long Ouyang, Jeffrey Wu, Xu Jiang, Diogo Almeida, Carroll Wainwright, Pamela Mishkin, Chong Zhang, Sandhini Agarwal, Katarina Slama, Alex Ray, et al. Training language models to follow instructions with human feedback. *Advances in Neural Information Processing Systems*, 35: 27730–27744, 2022.

Alec Radford, Jong Wook Kim, Chris Hallacy, Aditya Ramesh, Gabriel Goh, Sandhini Agarwal, Girish Sastry, Amanda Askell, Pamela Mishkin, Jack Clark, et al. Learning transferable visual models from natural language supervision. In *International Conference on Machine Learning*, pp. 8748–8763. PMLR, 2021.

Rafael Rafailov, Archit Sharma, Eric Mitchell, Stefano Ermon, Christopher D Manning, and Chelsea Finn. Direct preference optimization: Your language model is secretly a reward model. *arXiv preprint arXiv:2305.18290*, 2023.

Colin Raffel, Noam Shazeer, Adam Roberts, Katherine Lee, Sharan Narang, Michael Matena, Yanqi Zhou, Wei Li, , and Peter J. Liu. Exploring the limits of transfer learning with a unified text-to-text transformer. *Journal of Machine Learning Research*, 21:5485–5551, 2020.

Aditya Ramesh, Prafulla Dhariwal, Alex Nichol, Casey Chu, and Mark Chen. Hierarchical text-conditional image generation with clip latents. *arXiv preprint arXiv:2204.06125*, 2022.

Robin Rombach, Andreas Blattmann, Dominik Lorenz, Patrick Esser, and Björn Ommer. High-resolution image synthesis with latent diffusion models. In *Proceedings of the IEEE/CVF Conference on Computer Vision and Pattern Recognition*, pp. 10684–10695, 2022.

Nataniel Ruiz, Yuanzhen Li, Varun Jampani, Yael Pritch, Michael Rubinstein, and Kfir Aberman. Dreambooth: Fine tuning text-to-image diffusion models for subject-driven generation. In *Proceedings of the IEEE/CVF Conference on Computer Vision and Pattern Recognition*, pp. 22500–22510, 2023.

Chitwan Saharia, William Chan, Huiwen Chang, Chris A. Lee, Jonathan Ho, Tim Salimans, David J. Fleet, and Mohammad Norouzi. Palette: Image-to-image diffusion models. *arXiv preprint arXiv:2111.05826*, 2021a.

Chitwan Saharia, Jonathan Ho, William Chan, Tim Salimans, David J Fleet, and Mohammad Norouzi. Image super-resolution via iterative refinemen. *arXiv preprint arXiv:2104.07636*, 2021b.

Chitwan Saharia, William Chan, Saurabh Saxena, Lala Li, Jay Whang, Emily Denton, Seyed Kamyar Seyed Ghasemipour, Raphael Gontijo Lopes, Burcu Karagol Ayan, Tim Salimans, Jonathan Ho, David J. Fleet, and Mohammad Norouzi. Photorealistic text-to-image diffusion models with deep language understanding. In *Conference on Neural Information Processing Systems*, 2022.

John Schulman, Sergey Levine, Pieter Abbeel, Michael Jordan, and Philipp Moritz. Trust region policy optimization. In *International Conference on Machine Learning*, pp. 1889–1897. PMLR, 2015.

John Schulman, Filip Wolski, Prafulla Dhariwal, Alec Radford, and Oleg Klimov. Proximal policy optimization algorithms. *arXiv preprint arXiv:1707.06347*, 2017.

Kihyuk Sohn, Nataniel Ruiz, Kimin Lee, Daniel Castro Chin, Irina Blok, Huiwen Chang, Jarred Barber, Lu Jiang, Glenn Entis, Yuanzhen Li, et al. Styledrop: Text-to-image generation in any style. *arXiv preprint arXiv:2306.00983*, 2023.

Jiaming Song, Chenlin Meng, and Stefano Ermon. Denoising diffusion implicit models. *arXiv preprint arXiv:2010.02502*, 2020.

Yang Song and Stefano Ermon. Generative modeling by estimating gradients of the data distribution. In *Conference on Neural Information Processing Systems*, 2019.

Xiaoshi Wu, Keqiang Sun, Feng Zhu, Rui Zhao, and Hongsheng Li. Better aligning text-to-image models with human preference. *arXiv preprint arXiv:2303.14420*, 2023.

Jiazheng Xu, Xiao Liu, Yuchen Wu, Yuxuan Tong, Qinkai Li, Ming Ding, Jie Tang, and Yuxiao Dong. Imagereward: Learning and evaluating human preferences for text-to-image generation. *arXiv preprint arXiv:2304.05977*, 2023.

