# OpenReview forum: "Proximal Preference Optimization for Diffusion Models"
_ICLR.cc/2024/Conference — ICLR 2024 Conference Withdrawn Submission_

### Official Review · Reviewer_kXqb · 2023-10-29

**Soundness:** 2 fair
**Presentation:** 3 good
**Contribution:** 2 fair
**Rating:** 5
**Confidence:** 5

**Summary:**

The paper under review proposes a new approach named Proximal Preference Optimization for Diffusion Models (PPOD) to optimize the fine-tuning process in reinforcement learning with AI feedback. The proposed approach extends prior work on Direct Preference Optimization (DPO) by applying it to diffusion models often used in image-generative applications. The method involves the use of proximal constraints to meet optimization challenges that have been encountered when working with diffusion models. In comparison to existing optimization methods that largely rely on the use of reward optimization, PPOD focuses on preference optimization that is both stable and feasible within a supervised learning environment. The performance of PPOD was evaluated against other RL-based reward optimization approaches, leading to more impressive outcomes when applied to the stable diffusion model.

**Strengths:**

1. The paper introduces a new approach for fine-tuning diffusion models in the field of reinforcement learning. By leveraging the principles of Direct Preference Optimization (DPO) and introducing proximal constraints, the authors have provided a promising and innovative strategy for improving optimization in image-generative applications.
2. The paper is well-structured and clearly written.

**Weaknesses:**

1. A main weakness of the paper lies in its limited comparison with existing methods. While the paper evaluates PPOD against other RL-based approaches, it does not compare it with Denoising Diffusion Policy Optimization (DDPO)[1], which would have provided useful insights.
2. The novelty of the method appears to be moderate. The method largely builds on existing techniques within the field (i.e., the integration of DPO[2], diffusion models, and Image Reward Model[3] ). It could be viewed as a straightforward combination of these methods.
3. No code provided.

- [1] Black, Kevin, et al. "Training diffusion models with reinforcement learning." arXiv preprint arXiv:2305.13301 (2023).
- [2] Rafailov, Rafael, et al. "Direct preference optimization: Your language model is secretly a reward model." arXiv preprint arXiv:2305.18290 (2023).
- [3] Xu, Jiazheng, et al. "Imagereward: Learning and evaluating human preferences for text-to-image generation." arXiv preprint arXiv:2304.05977 (2023).

**Questions:**

1. Could you elaborate on the specific scenarios where PPOD might show superior performance compared to other RL-based reward optimization techniques?
2. Could you discuss any potential limitations of the method when applied within a supervised learning context? Are there any types of data or specific scenarios where it might not perform as expected?
3. Will you release your code in the future?

---

### Official Review · Reviewer_NEPt · 2023-10-30

**Soundness:** 2 fair
**Presentation:** 1 poor
**Contribution:** 1 poor
**Rating:** 3
**Confidence:** 3

**Summary:**

This paper adopts the Direct Preference Optimization (DPO) algorithm used for training the autoregressive language model and applies it to the training of a text-to-image diffusion model.

**Strengths:**

- The method is straightforward.
- The research is well-motivated.
- The related work section has a lot of details. Well done.

**Weaknesses:**

- This paper exhibits clear indications of last-minute preparations. The clarity of this paper can be significantly improved. More technical details should be given on how logits are computed, how images are sampled, etc.
- My own experience with ImageReward is that it is very sensitive to the initial Gaussian noise (x_T) which an image is sampled from. You can use the same reference model and the same text prompt to generate several images, but get a high variance in reward scores due to different starting x_T. Please consider including an analysis of the effect of random seeds.
- More metrics should be considered other than the ImageReward.
- The technical contribution is limited. This paper borrows the idea of the DPO algorithm and puts it into training diffusion models. Most of the derivation works were done in the DPO paper.

**Questions:**

Follow-up from weakness #1 (W1):
- imo, the main difficulty in transferring the fine-tuning techniques used for LLM to the fine-tuning diffusion model is defining the "policy". What is the state space and the action space? And why did you define the "last action" as a part of the current state? Does it make sense to have your action space similar to your state space? How did you handle the huge variance in continuous action space if it is a 64 x 64 tensor? I understand that this is a follow-up work from DPOK, but please consider explaining it clearly.
- How did you sample from a diffusion policy?
- Where did you get the $\pi_{ref}$ from? Is it a stable diffusion model defined under the diffusion policy?
- If no RL is involved, maybe you don't have to define a diffusion policy and compute the logits from a policy altogether. You can compute the exact likelihood of an image by integrating the ODE. Can you show a comparison between the two methods?
- In DPO, you can get the exact likelihood of generating a sequence from the autoregressive policy given it is well-defined, so we can be sure that the DPO is optimizing for the correct objective. But is it the case under the diffusion policy formulation?

---

### Official Review · Reviewer_MtxF · 2023-10-31

**Soundness:** 4 excellent
**Presentation:** 4 excellent
**Contribution:** 3 good
**Rating:** 8
**Confidence:** 5

**Summary:**

Inspired by progress in RLHF for text generation, this work incorporates Direct Preference Optimization (DPO) into the text-to-image diffusion setting. By generating preferences through model samples, this work shows the impressive capabilities of preference optimization for diffusion models as well.

**Strengths:**

- Clear application of DPO and ideas from recent work to generate preference data [1] to improve controllable text-to-image diffusion fine-tuning.
- Strong experimental results comparing against the growing literature of using RL for diffusion
- Compared different preference losses similar to the investigation being done recently for RLHF


[1] STATISTICAL REJECTION SAMPLING IMPROVES PREFERENCE OPTIMIZATION, Liu et al. 2023

**Weaknesses:**

Minor weakness:
- It would be interesting to see an extension from pairwise BT models to multiple rankings (Plakett-luce models) formulation. Given that the preferences come from the model itself, I wonder if there would be an improvement.

**Questions:**

see the question in the weaknesses section.

---

### Official Review · Reviewer_iw1V · 2023-11-03

**Soundness:** 1 poor
**Presentation:** 1 poor
**Contribution:** 2 fair
**Rating:** 3
**Confidence:** 5

**Summary:**

The authors present proximal preference optimization for diffusion models (PPOD), which extends direct preference optimization (DPO) to the text-to-image diffusion model setting. They introduce an online iterated version of DPO, as well as PPO-inspired ratio clipping, and show that both improve performance in the text-to-image setting. They perform experiments on Stable Diffusion 1.5 using ImageReward to obtain preferences and show good results on 4 prompts, as well as improved performance over prior work.

**Strengths:**

- The idea is simple and easy to follow, which is a strength in my book. Applying DPO to text-to-image diffusion models is a valuable avenue of research given its success with language models.
- The two modifications to DPO -- namely, online iteration and ratio clipping -- are interesting and fairly unexplored in the academic community, as far as I know. The experiments showing the importance of both decisions are convincing.

**Weaknesses:**

- In Section 3.2, the authors state: "Different from the language modeling, the KL divergence for diffusion models needs to regularize the whole denoising process between the finetuned and reference policies." This seems like a significant design decision that requires some explanation and justification. As a result of this decision, Equation (3) does not actually match the standard RLHF setup, due to the discrepancy between $\pi(\mathbf x_0 \mid \mathbf c)$ in the first term and $\pi(\mathbf x_{0:T} \mid \mathbf c)$ in the KL term.
- On a related note, the usage of a "policy" $\pi$ throughout the section is a bit confusing since it switches between $\pi(\mathbf x_0 \mid \mathbf c)$ and $\pi(\mathbf x_{0:T} \mid \mathbf c)$; furthermore, neither of these match the actual MDP policy defined in the referenced works DPOK and DDPO, which is $\pi(\mathbf x_{t-1} \mid \mathbf x_t, \mathbf c)$.
- Also related: this discrepancy between $\pi(\mathbf x_0 \mid \mathbf c)$ and $\pi(\mathbf x_{0:T} \mid \mathbf c)$ somewhat breaks the connection to DPO and its corresponding theoretical guarantees. Since the "policy" is considered to be $\pi(\mathbf x_{0:T} \mid \mathbf c)$ in the KL penalty, in order to match the DPO derivation, the ground-truth reward function must also be considered to operate on tuples $r(\mathbf x_{0:T}, \mathbf c)$. However, in reality, the reward function only depends on the final sample $\mathbf x_0$. Not sure what the downstream effects of this discrepancy would be, but it should be addressed by the authors.
- Algorithm 1 is quite confusing. For example, on the 3rd line, the reader must infer that the images are being sampled from the current model being optimized rather than $\pi_\text{ref}$. The explanation below it of offline vs. online learning is also quite confusing. This section could overall use some clarification, expansion, and introduction of more precise notation.
- The paper really emphasizes the term "RLAIF" throughout. Calling the method RLAIF seems a bit silly -- although there's no widely-accepted definition of "RLAIF", and I usually don't think quibbling over semantics is very productive, the fact that ImageReward is learned directly from human preferences means that using it as a reward model is not at all different from the "standard" RLHF setup. The authors also point out that their method is specifically *not* RL -- although that is also debatable, especially for the version where they perform multiple iterations of online improvement.
- The experiments are overall unconvincing. First, they only cover 4 prompts, and the model is optimized on a single prompt at a time. This severely limits practical applicability. Second, the primary comparison in Figure 3 does not really show a significant difference between DPOK and PPOD -- the small difference could be easily attributed to hyperparameter tuning. On top of that, it is well-known that reward functions (especially learned ones) are easily exploitable, and just because a method produces high-reward samples does not mean that those samples are desirable. Without any qualitative examples of the baselines, it is impossible to tell if the higher-reward samples from PPOD are actually better than the corresponding samples from DPOK. Third, the paper is severely lacking in hyperparameters and training details.
- One of the primary benefits of DPO is that it skips learning a reward model and learns directly from preference annotations instead. Using a learned reward model -- ImageReward -- as a preference annotator nullifies this benefit. PPOD is essentially doing RLHF, but adds an extra step of using the reward model to produce pairwise preference annotations and then uses those to optimize the model with the DPO objective. It is still possible that the DPO formulation improves optimization ability -- the DPO paper finds that their method can beat PPO even when PPO has access to the ground-truth rewards that generated the preference dataset. However, the authors do not make this the clear narrative of their paper.

Minor mistakes:
- Section 2: "clapping the gradient" instead of "clipping the gradient"
- Section 2: "The learning efficiency is not well optimized since they all required to build a reward model from the preference data." The works being referred to -- Black et. al. (2023) and Fan et. al. (2023) -- introduce generic RL algorithms that do not necessarily need to learn from preference data. In fact, none of the reward functions tested in Black et. al. (2023) are derived from pairwise preference data.
- Section 3.1: Define $\beta_t$ and $\sigma_t$.
- Algorithm 1: Inconsistency between superscript and subscript for $k$.
- Section 4.2: It is not mentioned whether it is the online or offline version of PPOD that is being reported in Figure 3.
- Table 1 should go before the references.

**Questions:**

- In the last sentence of the abstract, the authors write: "To the best of our knowledge, we are the first work that enabled the efficient optimization for the RLAIF on the diffusion models". Could they clarify what they mean by this statement? Two prior works that are cited in the introduction -- Black et. al. (2023) and Fan et. al. (2023) -- both perform RLAIF on diffusion models under the authors' definition. The latter even uses the same AI model, namely ImageReward.
- For the DPOK baseline: in Section 4.1, the authors state that they "follow the training setup described in the DPOK paper". However, in Section 4.2, they say that "The scores of SFT and DPOK are obtained from the DPOK paper". Did the authors re-implement DPOK, or simply take the ImageReward scores directly from the paper?
- Why did the authors not compare to DDPO, considering that the code for that paper has been released?
- Did the authors use LoRA or full finetuning?
- What is the reason that the authors did not attempt to apply PPOD directly to human preference data, which would much better fit the strengths of DPO?